# Supplemental Enzyme and Probiotics on the Growth Performance and Nutrient Digestibility of Broilers Fed with a Newly Harvested Corn Diet

**DOI:** 10.3390/ani12182381

**Published:** 2022-09-12

**Authors:** Caiwei Luo, Liqun Wang, Yanhong Chen, Jianmin Yuan

**Affiliations:** State Key Laboratory of Animal Nutrition, College of Animal Science and Technology, China Agricultural University, No. 2 Yuan Ming Yuan Western Road, Hai Dian District, Beijing 100193, China

**Keywords:** newly harvested corn, broiler, protease, glucoamylase, *Pediococcus acidilactici*

## Abstract

**Simple Summary:**

As an important energy feed material for livestock and poultry, corn is often in short supply and used immediately after harvest. However, newly harvested corn can have anti-nutritional factors (ANFs) and cause animal diarrhea. Feed-grade enzymes and probiotics can help to alleviate ANFs in feedstuffs. Therefore, in this study, several enzymes and a laboratory *Pediococcus acidilactici* BCC-1, isolated from chick ceca, were added into the newly harvested corn diet for a 21-day trial in broilers. Here, we found that the newly harvested corn diets resulted in shorter digesta emptying time, increased visual fecal water content, and decreased trypsin activity, which may lead to the occurrence of diarrhea in broilers. The supplementation of *Pediococcus acidilactici* BCC-1 to newly harvested corn diets could improve the activity of trypsin and decrease the FCR (F/G) of broilers. Moreover, supplemental protease could significantly increase the digesta emptying time, reduce the visual fecal water content, and increase trypsin activity, which may relieve the diarrhea of broilers. In conclusion, *Pediococcus acidilactici* BCC-1 or protease individually, as well as in combination with glucoamylase are recommended in newly harvested corn diets for broilers.

**Abstract:**

A new grain phenomenon happens in newly harvested corn because of its high content of anti-nutritional factors (ANFs), which can cause low nutrient digestibility and diarrhea in animals. Enzymes and probiotics have been shown to relieve the negative effect of ANFs for animals. The purpose of this study was to investigate the effect of enzymes and probiotics on the performance and nutrient digestibility of broilers, fed with newly harvested corn diets. A total of 624 Arbor Acres Plus male broiler chickens were randomly divided into eight treatment groups (A: normal corn diet, CT: newly harvested corn diet, DE: newly harvested corn diet + glucoamylase, PT: newly harvested corn diet + protease, XL: newly harvested corn diet + xylanase, BCC: newly harvested corn diet + *Pediococcus acidilactici* BCC-1, DE + PT: newly harvested corn diet + glucoamylase + protease, XL + BCC: newly harvested corn diet + xylanase + *Pediococcus acidilactici* BCC-1). Each group was divided into six replicates, with 13 birds each. On day 21, growth performance, nutrient digestibility, and digestive enzyme activity were measured. Compared with the normal corn diet (PC), the newly harvested corn diet (NC) produced shorter digesta emptying time (*p* = 0.015) and increased visual fecal water content (*p* = 0.002) of broilers, however, there was no effect on performance. Compared to the newly harvested corn diet (NC), supplemental enzyme of DE increased the activity of chymotrypsin (*p* = 0.016), however, no differences in the digestibility of three kinds of organic matter, digesta emptying time, visual fecal water content, or performance were found. Supplemental protease (PT) significantly increased digesta emptying time (*p* = 0.004) and decreased the activity of maltase (*p* = 0.007). However, it had no effect on the digestibility of three kinds of organic matter or the performance of broilers. Supplemental xylanase (XL) decreased the activity of amylase (*p* = 0.006) and maltase (*p* < 0.001); however, it had no effect on digesta emptying time, visual fecal water content, the digestibility of three kinds of organic matter, or performance of broilers. Supplemental DE, combined with PT (DE + PT), increased the digesta emptying time (*p* = 0.016) while decreasing the visual fecal water content (*p* = 0.011), and the activity of amylase (*p* = 0.011), lipase (*p* = 0.021), and maltase (*p* < 0.001), however, there was no effect on performance. Supplemental BCC individually decreased the activity of amylase (*p* = 0.024) and maltase (*p* < 0.001), however, it increased the activity of trypsin (*p* < 0.001) and tended to improve feed conversion ratio (FCR) (*p* = 0.081). Supplemental BCC-1, combined with XL (XL + BCC), increased the activity of trypsin (*p* = 0.001) but decreased the activity of amylase (*p* = 0.013), lipase (*p* = 0.019), and maltase (*p* < 0.001). *Pediococcus acidilactici* BCC-1 (10^9^ cfu/kg), protease (800,000 U/g) individually, or protease (800,000 U/g) in combination with glucoamylase (800,000 U/g) were supplemented in newly harvested corn diets for growing broilers. Hence, this study mainly explores the alleviation effect of enzyme and probiotics on the negative phenomenon caused by the utilization of newly harvested corn in broilers and provides a better solution for the utilization of newly harvested corn in production practice.

## 1. Introduction

Newly harvested corn can cause diarrhea in livestock, including commercial poultry, and though the mechanism causing this diarrhea is not clear, it surely has an economic impact. Yin et al. (2022) [1] found that starch and polysaccharide biosynthesis in newly harvested corn had not completely finished, and higher levels of soluble sugars, which increased viscoelastic properties of corn and the relative digesta viscosity in chicken, can cause harmful bacteria, such as *Hungatella hathewayi* and *Bacteroides fragilis*, in the gut of broiler chickens, leading to disturbance of intestinal microflora and impaired absorption and digestion.

There are many anti-nutritional factors in corn. Cornstarch is composed of amylose (AM) and amylopectin (AP), and amylopectin not only contains α-1, 4-glucosidic bonds but also α-1, 6-glucosidic bonds. Animal pancreases only secrete α-amylase to degrade α-1, 4-glucosidic bonds, while amylopectin can only be hydrolyzed by amylopectase [2]. Corn has a high concentration of insoluble non-starch polysaccharides (NSP), such as xylan and cellulose [3], which limit the ability of digestive enzymes to access and fully digest the starch and protein components enclosed within the plant cells. Additionally, large amounts of native trypsin inhibitors (0.56 and 1.87 mg/g dry matter (DM)) were found in corn [4].

Utilizing supplemental enzymes to eliminate the negative effects of anti-nutritional factors of feed ingredients is the common practice [5]. Dietary supplemental xylanase individually could decrease intestinal digesta viscosity, inhibit pathogenic microorganisms, improve the absorption capacity of intestine so as to increase the digestibility of nutrient, and improve the performance of broilers [6]. Amylase could increase the digestibility of starch and AME and decrease the feed conversion ratio (FCR) of broilers fed with corn-soybean meal diets, however, different sources and concentrations of amylase had varying effects on broilers [7]. Exogenous α-amylase impacted starch digestibility in the jejunum and ileum and influenced gut microbiota [8]. Protease could change the intestinal microbial richness and diversity in broiler chickens [9]. However, Yegani and Korver (2013) [10] investigated the effects of exogenous enzymes (xylanase, xylanase combined with amylase and protease, or xylanase combine with β-glucanase) on growth performance, ileal digestible energy (IDE), and apparent ileal digestibility of CP and amino acids (AA) in broiler chickens fed corn soybean meal diets, formulated using three different sources of corn. These authors found that the effects of enzyme products on IDE and digestibility of CP and AA were not consistent and varied depending on corn sources, enzyme products, and dietary phases. Even with similar proximate compositions, different corn sources had different salt-soluble protein content, which may be used to differentiate those with wide-ranging IDE or AMEn [11]. Yin et al. (2018) [12] found supplemental amylase, combined with glucoamylase or protease in newly harvested corn produced a beneficial effect on starch digestibility and intestinal microbiota diversity, while increasing the growth of broilers fed with newly harvested corn. However, supplemented α-amylase, combined with amylopectase, glucoamylase, protease, and xylanase, had a tendency to decrease body weight gain of broilers, and increase the count of *Campylobacter*, *Helicobacter* and *Butyricicoccus* in the ceca. However, the role of protease and xylanase in newly harvested corn is not clear.

Probiotics are a class of active microorganisms that are believed to improve gut micro ecology, regulate intestinal permeability, and improve the nutrient digestion and absorption, thereby ameliorating its dysfunction-associated pathological conditions [13,14,15,16]. At present, increasing studies have suggested that probiotics, such as *Lactobacillus* and *Bacillus*, have a positive effect on the broiler growth and nutrient digestion [17,18,19,20]. In the previous research, we isolated a strain *Pediococcus acidilactici* BCC-1 from the ceca of broilers fed a wheat-based diet, supplemented with xylanase, and found it has the propensity to utilize XOS efficiently [21]. *Pediococcus acidilactici* belongs to *Lactobacillaceae* and is permitted to be used as a feed additive in many countries. As far as we are aware, whether *Pediococcus acidilactici* BCC-1 individually or in combination with enzymes can utilize the high level of soluble sugars in newly harvested corn, promote broiler growth, and improve nutrient digestion and absorption in newly harvested corn diet is unknown. Therefore, in this study, xylanase, protease, amylase and probiotics individually, or in combination, will be used in the diets of broilers fed with newly harvested corn. This study will try to explore the effect of supplementation of enzymes and probiotics on the growth performance and nutrient digestibility of broilers fed with newly harvested corn diets and provide a better solution for the utilization of newly harvested corn in production practice and a theoretical basis and scientific reference for alleviating the shortage of corn resources in the poultry industry.

## 2. Materials and Methods

All animal procedures were conducted in accordance with the Beijing Regulations of Laboratory Animals (Beijing, China) and were approved by the Laboratory Animal Ethical Committee of China Agricultural University (Protocol Number: AW03602202-1-3).

### 2.1. Experimental Diets and Treatments

A total of 624 one-day-old male *Arbor Acres Plus* broiler chickens (from Beijing Poultry Breeding Company, Beijing, China) were randomly divided into 8 treatment groups, each group including 6 replicate cages, with 13 birds each. There were 2 control treatments, normal corn (stored for one year) as the positive control (PC), and newly harvested corn (just harvested within one half month) as the negative control (NC). The specific treatment of the experiment group is shown in Table 1. The enzymes were all purchased from Beijing Huameiyuan Biotechnology Co., Ltd. (Beijing, China). *Pediococcus acidilactici* BCC-1 was isolated by the lab, and 10^9^ cfu/kg was added to the diet.

The newly harvested corn and normal corn were collected from Zhuozhou (Hebei, China); its chemical composition analysis is shown in Table 2. Broiler chickens were fed a corn soybean meal basal diet, as shown in Table 3.

The trial was performed in the Zhuozhou Poultry Research Base of China Agricultural University (Beijing, China) in Autumn. The initial temperature of the chicken house was set to 33 °C, and then gradually decreased to 24 °C on day 21. All birds had access to feed and water ad libitum in crumble pellet form and via nipple drinkers, respectively. Bird management was based on the guide of *Arbor Acres Plus* broilers.

### 2.2. Feed Passage Rate

On day 18, feed was withdrawn from each pen for 2 h and identical diets containing chromic oxide (1.0 g/kg) were then offered for 15 min before being replaced with the original diet. Transit time was determined as the lapsed time from the introduction of the diets to the time of first appearance in the droppings of green coloration from chromic oxide [22]. The excreta were collected for moisture determination.

### 2.3. Performance Data Collection

On day 21, after fasting for 12 h, feed intake (FI) and body weight (BW) of broilers were measured on a cage-by-cage basis, and body weight gain (BWG) and FCR were calculated. Mortality was recorded daily.

### 2.4. Metabolic Trial

After fasting for 6 h on day 21, the birds were fed a test diet supplemented with 0.5% TiO_2_. After being re-fed for 3 h, three birds with weight close to the average body weight in each replicate were selected, and the birds were euthanized by intravenous injection of sodium pentobarbital (30 mg/kg). Terminal ileal digesta were collected and mixed for analyzing the digestibility of crude protein, crude fat and starch. The jejunal digesta was collected and jejunal mucosa was scraped, snap-frozen in liquid nitrogen immediately, and stored at −80 °C for digestive enzyme assay.

### 2.5. Chemical Analysis

The ileal digesta samples were subsequently freeze-dried and ground to pass through a 40 µm strainer before analyses. The diet was ground to pass through the 40 µm strainer also. Analysis of the indigestible marker, titanium dioxide (TiO_2_), was conducted using the method proposed by Short et al. (1996) [23] and with minor modifications according to Myers et al. (2004) [24]. The nitrogen in the digesta and diet was determined by the Kjeldahl method (FOSS KT 200 Kjeltec, Hillerod, Denmark). The crude fat in the digesta and diet was determined by the Soxtec extraction procedure [25]. The starch in the digesta and diet was determined using a Megazyme kit. The moisture content of dropping was determined by drying at 105 °C for 2 h.

The apparent ileal digestibility (*AID*) of crude protein, crude fat, and starch was calculated using the following formula:(1)$$AID,\%=\left[1-\left(\frac{\%\;Nutrient_{digesta}}{\%\;Nutrient_{diet}}\right)\times \left(\frac{\%\;Marker_{diet}}{\%\;Marker_{digesta}}\right)\right]\times 100$$

### 2.6. Digestive Enzyme Assay

The jejunal digesta samples were centrifuged at 1500× *g* at 4 °C for 10 min. The supernatants were used to determine activities of amylase, chymotrypsin, and trypsin according to the method of a commercially available assay kit (Jiancheng Bioengineering Institute, Nanjing, China).

The jejunal mucosa samples were homogenized and centrifuged at 1500× *g* at 4 °C for 10 min, and then the supernatants were harvested for sucrase and maltase analyses according to the commercially available assay kit manufacturer’s recommendation (Jiancheng Bioengineering Institute, Nanjing, China). The protein content of the above samples was determined by using BCA protein assay kit according to the manufacturer’s recommendation (Jiancheng Bioengineering Institute, Nanjing, China).

### 2.7. Statistical Analysis

Data were first tested for homogeneity of variances, and then analyzed by one-way analysis of variance (ANOVA), followed by post hoc analysis using Duncan’s multiple comparison tests, using SPSS 20.0 statistical software (version 20.0, SPSS Inc., USA). Growth performance and feed passage rate data were based on the cage (replicate) as the experimental unit. For other experimental data (nutrient digestibility and digestive enzyme activity), one bird in each cage was selected as the experimental unit. The statistical model for this study is shown below. All data values are represented as the mean and SEM. Differences between the treatment groups were considered statistically different at *p* < 0.05.
$$Y_{ij}=\mu +T_{i}+\varepsilon_{ij}$$
where: *Y_ij_* = a single observation, μ = overall mean, $T_{i}$ = effect of diet (normal or newly harvested corn diet and supplemental different enzymes and probiotics), $\varepsilon_{ij}$ = the experimental random error.

## 3. Results

### 3.1. Growth Performance

There was no differences for the performance of broilers between normal corn diet (PC) and newly harvested corn diet (NC) (*p* > 0.05) (Table 4). Compared with the NC, the supplementation of different enzymes or enzymes combined with probiotics did not affect BW or BWG of broilers (*p* > 0.05), however, supplemental DE tended to increase FI (*p* = 0.075), and supplemental BCC (*p* = 0.073) individually or in combination with xylanase (XL + BCC) (*p* = 0.054) tended to decrease FI. These translated into tendencies to increase FCR for supplemental glucoamylase (DE) (*p* = 0.096) and xylanase (XL) (*p* = 0.085), and a tendency to decrease FCR of broilers for supplemental BCC (*p* = 0.081). Compared with supplemental enzymes DE, PT, XL, and DE + PT, supplemental BCC (*p* = 0.001, 0.015, 0.003, and 0.004, respectively) individually, or combined with xylanase (XL + BCC) (*p* = 0.000, 0.011, 0.002 and 0.003, respectively) significantly decreased FI.

### 3.2. Digesta Emptying Time and Visual Fecal Water Content Determination

Diarrhea was preliminarily judged by the digesta emptying time and visual fecal water content. Compared with the normal corn diet (PC), the newly harvested corn diet (NC) produced shorter digesta emptying time (*p* = 0.015) and increased visual fecal water content (*p* = 0.002) of broilers (Table 5).

Compared with the NC, supplemental protease (PT) (*p* = 0.004) or protease combined with glucoamylase (DE + PT) (*p* = 0.016) increased the digesta emptying time, and supplemental protease combined with glucoamylase (DE + PT) (*p* = 0.011) decreased the visual fecal water content. Moreover, supplemental glucoamylase (DE), xylanase (XL) and BCC individually or in combination with xylanase (XL + BCC) had no significant effect on digesta emptying time and visual fecal water content (*p* > 0.05).

### 3.3. Nutrient Digestibility

There was no difference for the digestibility of crude protein or crude fiber between the newly harvested corn diet (NC) and normal corn diet (PC) (*p* > 0.05) (Table 6). However, newly harvested corn (NC) tended to decrease the digestibility of starch, compared to normal corn diet (PC) (*p* = 0.082).

Supplemental enzymes and probiotics (DE, PT, XL, BCC, DE + PT, and XL + BCC) in newly harvested corn diet did not affect the digestibility of crude protein, crude fiber, or starch in broilers (*p* > 0.05), compared with the NC. However, supplemental protease (PT) in a newly harvested corn diet significantly decreased the digestibility of starch, compared to the NC (*p* = 0.035) or PC (*p* = 0.001). It is also worth mentioning that in terms of starch digestibility, the effect of the NC diets supplemented with BCC and XL + BCC were comparable with the PC.

### 3.4. Digestive Enzyme Activity

There was no significant difference for activities of various digestive enzymes between the newly harvested corn group (NC) and normal corn group (PC) (*p* > 0.05) (Table 7). However, compared with the NC, supplemental glucoamylase (DE) significantly increased the activity of jejunal digesta chymotrypsin (*p* = 0.016) and supplemental protease (PT) significantly decreased the activity of jejunal digesta maltase (*p* = 0.007). Supplemental xylanase (XL) significantly decreased the activity of jejunal digesta amylase (*p* = 0.006) and maltase (*p* < 0.001). Supplemental glucoamylase combined with protease (DE + PT) significantly decreased the activity of jejunal digesta amylase (*p* = 0.011), lipase (*p* = 0.021), and maltase (*p* < 0.001). Supplemental BCC individually significantly increased the activity of jejunal digesta trypsin (*p* < 0.001) and decreased the activity of jejunal digesta amylase (*p* = 0.024) and maltase (*p* < 0.001). Supplemental BCC, combined with xylanase (XL + BCC), significantly increased the activity of jejunal digesta trypsin (*p* = 0.001) but decreased the activity of jejunal digesta amylase (*p* = 0.013), lipase (*p* = 0.019), and maltase (*p* < 0.001). No significant difference was detected in the activity of jejunal mucosal sucrase among the groups (*p* > 0.05).

## 4. Discussion

Newly harvested corn contains a variety of ANFs, such as phytic acid, NSP, and soybean antigenic protein, which may reduce the bioavailability of nutrients [26] and increase viscoelastic properties of corn and the relative digesta viscosity [1], resulting in intestinal disorders, which, in turn, may lead to the occurrence of diarrhea in poultry. In addition, corn contains approximately 0.56–1.87 mg/g (DM) [4] native trypsin inhibitors, which may limit the contact of digestive enzymes with substrates in the intestinal tract of broilers [12]. This can result in impaired nutrient transport and/or digestion and greatly increased water intake. This provides more opportunities for the growth of harmful bacteria, such as *Clostridium perfringens*, in the intestinal tract [27], which can adversely affect the intestinal health of broilers. Therefore, it is urgent to solve the nutritional problems brought by newly harvested corn.

Poultry diarrhea is mainly characterized by loose stools with high moisture content and occasional undigested feed or pus and blood [28,29]. Therefore, it can be preliminarily judged whether the broilers have diarrhea by the digesta emptying time and the visual fecal water content. The results of this study showed that broiler chickens fed newly harvested corn diets significantly decreased their digesta emptying time and increased visual fecal water content. Furthermore, during the test, the undigested feed was observed, which proved that there are indigestible components in newly harvested corn and feeding the newly harvested corn diet would cause diarrhea in broilers, this is consistent with the findings of Yin et al. (2022) [1]. However, this phenomenon was alleviated by the supplementation of protease individually or in combination with glucoamylase in newly harvested corn diets. A possible explanation for this might be that proteases alter the diet starch–protein interface, affecting the rate of nutrient digestion and absorption, and enhancing intestinal reabsorption of water [12], but the specific reasons remain to be further studied.

Recently, there is accumulating evidence demonstrating that there are considerable beneficial health effects of enzymes and probiotics in terms of efficiency to enhance poultry growth, improve intestinal health, and suppress pathogens [30,31,32,33]. Interestingly, our research shows that there were no significant differences in broilers BW and BWG and nutrient digestibility between supplementation of enzymes and probiotics, compared with the newly harvested corn diet group. This is consistent with the findings of Hussein et al. (2020) [34], Al-Khalaifa et al. (2019) [18], and Tari et al. (2022) [35]. This result may be related to the good balance of digestible amino acids and the high inherent protein digestibility in the basal diet and complex changes in digestive enzyme activities. Surprisingly, we found that the supplementation of protease to the newly harvested corn diet significantly decreased starch digestibility, compared with the normal corn diet. Several possible factors could explain this observation. Firstly, it is possible that the supplementation of protease ameliorated the phenomenon of diarrhea in broilers, slowed the intestinal digesta transport, and delayed the movement of indirect markers [36]. Secondly, the supplementation of exogenous protease could interfere with the secretion of endogenous digestive enzymes in broilers, which may lead to negative feedback regulation. Another finding of this study is that feeding broilers with normal corn is not significantly better than newly harvested corn, which may be due to the differences in nutrients between the two corn varieties. Moreover, the FCR in the supplementation of *Pediococcus acidilactici* BCC-1 group tended to be decreased, compared with the newly harvested corn diet group. This is similar to the previous findings of broilers [17,37], and this result may be explained by the fact that, compared with the newly harvested corn diet group, *Pediococcus acidilactici* BCC-1 can numerically improve nutrient digestibility, albeit without a significant difference, which may also be the reason why *Pediococcus acidilactici* BCC-1 has a trend of increasing FCR but not significantly.

The small intestine of poultry is the major digestion and absorption site of nutrients, such as carbohydrates, proteins, and fats [38]. The digestive enzyme activity in intestinal digesta and mucosa is the main limiting factor affecting the FI, digestion, absorption, and growth of poultry [39,40]. Currently, there are various reports on the effects of probiotics on intestinal digestive enzymes. Zhang et al. (2016) [41] reported that the supplementation of *Clostridium butyricum* to the diet promoted the amylase, protease, and lipase activities in broilers challenged with *Escherchia coli K88*. A study by Rodjan et al. (2018) [42] showed that probiotics had no significant effect on the intestinal digestive enzyme activity of broilers. However, the results of this study indicated that adding enzymes and probiotics to the newly harvested corn diet could increase the activities of chymotrypsin and trypsin to varying degrees while decreasing the activities of amylase, lipase, and maltase. This finding was also reported by Jiang et al. (2020) [43] but is in contrast to the study by Gong et al. (2018) [44]. Studies by Gong et al. have shown that adding probiotics to the basal diets can significantly increase amylase and lipase activities. This rather contradictory result may be due to differences in the ability of different probiotic strains to produce enzymes and/or stimulate the endogenous enzymes production in broilers [45]. Another explanation could be differences in the pH within the gastrointestinal tract. It is well known that the pH in the gastrointestinal tract has important implications for nutrient absorption and gut microbiome in birds [46]. Only when its pH is maintained in a certain range can the activity of various digestive enzymes be ensured, so as to exert their digestion and absorption functions.

## 5. Conclusions

In summary, the newly harvested corn diet resulted in shorter digesta emptying time, increased visual fecal water content, and decreased trypsin activity, which may lead to the occurrence of diarrhea in broilers; however, it did not affect broiler growth performance and digestibility of crude protein, crude fiber, and starch. The supplementation of *Pediococcus acidilactici* BCC-1 to newly harvested corn diets improved broilers FCR and increased trypsin activity. Furthermore, the supplementation of protease or in combination with glucoamylase can significantly increase the digesta emptying time, reduce the visual fecal water content, and increase trypsin activity, which may alleviate the diarrhea of broilers. The specific reasons driving this are unknown, so further studies on the effects of enzymes and probiotics on the intestinal health of broilers fed with the newly harvested corn diet will be a fruitful direction for future research. Finally, we recommend supplementing *Pediococcus acidilactici* BCC-1 (10^9^ cfu/kg) or protease (800,000 U/g) individually, or protease (800,000 U/g) in combination with glucoamylase (800,000 U/g) in newly harvested corn diets for growing broilers.

## Figures and Tables

**Table 1 animals-12-02381-t001:** Treatment of the experiment.

Item	Treatment
PC	Normal corn (stored for one year) diet
NC	Newly harvested corn (just harvested within one half month) diet
DE	NC supplemented with α-amylase (*Aspergillus oryzae*) 900 U/kg and amylopectase (*Bacillus amyloliquofaciens*) 600 U/kg, glucoamylase (*Aspergillus usamil*) 60,000 U/kg
PT	NC supplemented with protease (*Bacillus subtilis*) 12,000 U/kg
XL	NC supplemented with xylanase (*Trichoderma longibrachiatum*)4,800 U/kg
BCC	NC supplemented with *Pediococcus acidilactici BCC-1* 10^9^ cfu/kg
DE + PT	NC supplemented with α-amylase (*Aspergillus oryzae*) 900 U/kg, amylopectase (*Bacillus amyloliquofaciens*) 600 U/kg, glucoamylase (*Aspergillus usamil*) 6,0000 U/kg, and protease (*Bacillus subtilis*) 12,000 U/kg
XL + BCC	NC supplemented *Pediococcus acidilactici BCC-1* 10^9^ cfu/kg combined with xylanase (*Trichoderma longibrachiatum*) 48,000 U/kg

**Table 2 animals-12-02381-t002:** Chemical composition analysis of normal corn and newly harvested corn.

Item	Normal Corn	Newly Harvested Corn
CP, %	9.32	9.20
Ash, %	1.14	1.25
Crude fat, %	3.50	2.83
Crude fibre, %	3.03	1.42
DM, %	88.86	87.3
Water-soluble pentosan, %	0.52	0.83
Total pentosan, %	3.94	5.27

Abbreviation: CP, crude protein; DM, dry matter.

**Table 3 animals-12-02381-t003:** Ingredient and nutrient composition of experimental diet (%, as-fed basis).

Item	Composition	Nutrient Levels	Composition	
Corn	57.00		Normal corn diet	Newly harvested corn diet
Soybean meal	35.75	Metabolizable energy (Mcal/kg) ^3^	2.91	
Soybean oil	2.85	CP, % ^4^	21.05	20.93
Dicalcium phosphate	2.00	Lysine, % ^3^	1.25	
Limestone	1.22	Methionine, % ^3^	0.57	
Sodium chloride	0.35	Methionine + cystine, % ^3^	0.91	
L-Lysine hydrochloride (78%)	0.12	Threonine, % ^3^	0.80	
DL-Methionine (98%)	0.26	Calcium, % ^3^	1.00	
Choline chloride (50%)	0.20	NPP, % ^3^	0.47	
Mineral premix ^1^	0.20	Starch, % ^4^	28.62	28.36
Vitamins premix ^2^	0.03	Crude fat, % ^4^	4.15	3.53
Antioxidant	0.02	DM, % ^4^	92.33	88.65
Total	100.00			

Abbreviation: NPP, Non-phytate phosphorus. ^1^ The trace mineral premix provided the following per kg of diets: Cu, 16 mg; Zn, 110 mg; Fe, 80 mg; Mn, 120 mg; Se, 0.30 mg; I, 1.50 mg. ^2^ The vitamin premix provided the following per kg of diets: vitamin A, 15,000 IU, vitamin D_3_, 3600 IU; vitamin E, 30 IU; vitamin K_3_, 3.00 mg; vitamin B_2_, 9.60 mg; vitamin B_12_, 0.03 mg; biotin, 0.15 mg; folic acid, 1.50 mg; pantothenic acid, 13.80 mg; nicotinic acid, 45 mg. ^3^ Calculated values of nutrient components. ^4^ Analysis values of nutrient components.

**Table 4 animals-12-02381-t004:** Supplemental enzymes with probiotics on growth performance of broiler chickens from 1 to 21 days.

Item	BW, g/Bird	BWG, g/Bird	FI, g/Bird	FCR
PC	543.14	409.32	671.20 ^ab^	1.65 ^a^
NC	570.46	435.03	671.86 ^ab^	1.59 ^ab^
DE	560.48	425.13	698.07 ^a^	1.65 ^a^
PT	573.78	439.71	681.78 ^a^	1.55 ^ab^
XL	560.77	425.77	690.71 ^a^	1.65 ^a^
BCC	563.63	428.58	645.48 ^b^	1.51 ^b^
DE + PT	570.90	436.01	688.91 ^a^	1.58 ^ab^
XL + BCC	548.60	414.98	643.40 ^b^	1.56 ^ab^
SEM	3.739	3.676	4.315	0.013
*p* value	0.404	0.428	0.002	0.065

Abbreviation: BW, body weight; BWG, body weight gain; FI, feed intake; FCR, feed conversion ratio. ^a,b^ Means in the same row with different superscripts indicate tended to differ or differ significantly (*p* < 0.05).

**Table 5 animals-12-02381-t005:** Supplemental enzymes with probiotics on digesta emptying time and visual fecal water content.

Item	Digesta Emptying Time, min	Visual Fecal Water Content, %
PC	143.05 ^a^	77.66 ^c^
NC	124.00 ^bc^	80.97 ^a^
DE	125.56 ^bc^	80.30 ^ab^
PT	147.17 ^a^	79.39 ^abc^
XL	123.81 ^bc^	80.19 ^ab^
BCC	117.79 ^c^	79.91 ^ab^
DE + PT	142.80 ^a^	78.25 ^bc^
XL + BCC	135.59 ^ab^	81.22 ^a^
SEM	42.148	0.290
*p* value	0.001	0.012

^a,b,c^ Means in the same row with different superscripts indicate tended to differ or differ significantly (*p* < 0.05).

**Table 6 animals-12-02381-t006:** Supplemental enzymes with probiotics on nutrient digestibility of broiler chickens at 21 d.

Item	Crude Protein, %	Starch, %	Crude Fat, %
PC	61.48	94.57 ^a^	60.16
NC	66.65	91.46 ^ab^	53.70
DE	65.25	90.46 ^ab^	51.22
PT	53.41	87.17 ^b^	47.98
XL	56.73	91.25 ^ab^	52.65
BCC	68.72	93.18 ^a^	57.22
DE + PT	60.11	91.21 ^ab^	60.43
XL + BCC	70.86	92.96 ^a^	51.71
SEM	1.619	0.545	1.768
*p* value	0.081	0.029	0.616

^a,b^ Means in the same row with different superscripts indicate tended to differ or differ significantly (*p* < 0.05).

**Table 7 animals-12-02381-t007:** Supplemental enzymes with probiotics on digestive enzyme activity of broiler chickens at 21 d.

Item	Chymotrypsin, U/mg Protein	Trypsin, U/mg Protein	Amylase, U/mg Protein	Lipase, U/mg Protein	Maltase, U/mg Protein	Sucrase, U/mg Protein
PC	24.54 ^ab^	437.10 ^c^	53.59 ^a^	45.52 ^ab^	175.28 ^ab^	64.04
NC	21.87 ^bc^	356.98 ^c^	48.03 ^a^	77.84 ^a^	207.805 ^a^	54.87
DE	29.23 ^a^	570.95 ^bc^	47.26 ^a^	69.40 ^ab^	159.73 ^ab^	56.51
PT	19.35 ^bc^	648.85 ^abc^	46.09 ^a^	70.14 ^ab^	139.19 ^b^	45.23
XL	20.38 ^bc^	568.56 ^bc^	27.39 ^b^	49.50 ^ab^	83.96 ^c^	58.47
BCC	24.46 ^ab^	912.69 ^a^	31.47 ^b^	34.95 ^ab^	81.33 ^c^	55.16
DE + PT	19.30 ^bc^	604.35 ^bc^	29.09 ^b^	32.06 ^b^	77.50 ^c^	47.82
XL + BCC	17.26 ^c^	845.27 ^ab^	29.58 ^b^	31.04 ^b^	66.95 ^c^	68.30
SEM	0.853	407.311	2.183	5.100	2.712	9.309
*p* value	0.005	0.004	0.001	0.090	<0.001	0.456

^a,b,c^ Means in the same row with different superscripts indicate tended to differ or differ significantly (*p* < 0.05).

## Data Availability

The data presented in this study are available on request from the corresponding author.

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
