# Peer review of "Supplemental Enzyme and Probiotics on the Growth Performance and Nutrient Digestibility of Broilers Fed with a Newly Harvested Corn Diet"

_animals, 2022, doi:10.3390/ani12182381_

Round 1

Reviewer 1 Report

This study of Luo et al. studied the impact of the effect of enzymes and probiotics on the performance and nutrient digestibility of broilers fed with newly harvested corn diets using 624 Arbor Acres Plus male broiler chickens that were randomly divided into 8 treatment groups (A: normal corn diet, CT: newly harvested corn diet, DE: newly harvested corn diet + glucoamylase, PT: newly harvested corn diet + protease, XL: newly harvested corn diet + xylanase, BCC: newly harvested corn diet + Pediococcus acidilactici BCC-1, DE+PT: newly harvested corn diet + glucoamylase + protease, XL+BCC: newly harvested corn diet + xylanase + Pediococcus acidilactici BCC-1). Each group was divided into 6 replicates, with 13 birds each. The authors concluded that ediococcus acidilactici BCC-1 and protease individually or in combination with glucoamylase are recommended in newly harvested corn diets for growing broilers. The use of enzymes and probiotics in poultry diets has been initiated 50 years with many successfully results and practical application to animal diets, thus the novelty of this study must be emphasis  before acceptance  and here is my comments  which could be summarized in the following:

1.  In the last  statement of the abstract,  plz supply the level of probiotics and enzymes for recommendation.

2. In the introduction section, please emphasis on the added value/novelty of this research and how it could contribute to improve production of yellow feathered broilers.

3.  Reference that could be of added value to your Ms in L 39:

- Al-Sagan Ahmed A., Abdullah H. AL-Yemni, Abdulaziz A. Al-Abdullatif, Youssef A. Attia and Elsayed O. S Hussein (2020).  Effects of different dietary levels of blue lupine (Lupinus angustifolius) seed meal with or without probiotics on the performance, carcass criteria, immune organs, and gut morphology of broiler chickens.   Frontiers Vet Sci  2020, 7, 124. Doi:10.3389/fvets.2020.00124

4. L 134, what form of feed did the chickens offered?

5. The statistical model and the experiment unit should be added to the statistical analyses section, L 193-195, you may also use Tukey post hock instead of Duncan Test.

6.  L 217, Plz adjust as and the visual fecal water content.

7.  L 329-339,  plz add supplemented dose of probiotics and enzymes for taken home massage 

Author Response

Response to Reviewer 1 Comments

Dear reviewers,

Thank you for the valuable comments from the reviewers concerning our manuscript entitled “Supplemental enzyme and probiotics on the growth performance and nutrient digestibility of broilers fed with a newly harvested corn diet” (Manuscript ID: animals-1848559). The comments have been highly valuable and helpful for revising and improving our manuscript. We have endeavored to incorporate the feedback and revised our manuscript accordingly, and use the “Track Changes” function for any revisions to the manuscript. The detailed answers to the reviewers are as follows:

Reviewer 1 Comments:

  • Q:In the last statement of the abstract, plz supply the level of probiotics and enzymes for recommendation.

Answer: Thanks for your comment. We have added the recommended dosage to the corresponding content. Please see L 52-55.

  • Q: In the introduction section, please emphasis on the added value/novelty of this research and how it could contribute to improve production of yellow feathered broilers.

Answer: Thanks for your comment. We have added the corresponding content. Please see L 112-116 and 120-122.

  • Q:Reference that could be of added value to your Ms in L 39:

- Al-Sagan Ahmed A., Abdullah H. AL-Yemni, Abdulaziz A. Al-Abdullatif, Youssef A. Attia and Elsayed O. S Hussein (2020).  Effects of different dietary levels of blue lupine (Lupinus angustifolius) seed meal with or without probiotics on the performance, carcass criteria, immune organs, and gut morphology of broiler chickens.   Frontiers Vet Sci  2020, 7, 124. Doi:10.3389/fvets.2020.00124

Answer: Thank you for your comment. We had read above paper and cited it to enrich our paper. Please see L 408.

  • Q:L 134, what form of feed did the chickens offered?

Answer: Thanks for your suggestion. We had revised the writing in our paper as advised. Please see L 153.

  • Q: The statistical model and the experiment unit should be added to the statistical analyses section, L 193-195, you may also use Tukey post hock instead of Duncan Test.

Answer: Thanks for your kind advice. We have added relevant content to the description of 2.7 Statistical Analysis. Please see L 202-211. In addition, we used the Tukey post hock to process the experimental data, and found that the results were not much different from the Duncan Test results, so we decided to use the Duncan Test to process the test data. Thank you very much for your valuable suggestions.

  • Q:L 217, Plz adjust as and the visual fecal water content.

Answer: Thanks for your comment and kind advice. We had already modified the writing. Please see L 230-242.

  • Q:L 329-339, plz add supplemented dose of probiotics and enzymes for taken home massage

Answer: Thanks for your comment. We have added the recommended dosage to the corresponding content. Please see L 360-362.

Please refer to the attachment for revised manuscript.

Finally, thank you again for your valuable suggestions for this study.

Best regards.

Reviewer 2 Report

The manuscript entitled 'Supplemental enzyme and probiotics on the growth performance and nutrient digestibility of broilers fed with a newly harvested corn diet' is quite novel and presents some new information in the use of enzymes and probiotics in poultry. There are a few minor issues that can be addressed below. 

I suggest the same parameters be given from Table 1 and 3. The information should be consistent for comparison of the corn (various strains) and the corn diets. Table 3 should include values for CF%, Ash%, soluble penostans. Starch value can also be placed in Table 1 if available.

In text referencing should be in the journal format (please revise throughout the manuscript) eg. Line 156, 319.

In the results section (3.3). Some mention can be made on the effect of the NC diets supplemented with BCC and XL+BCC were comparable with the PC with respect to starch digestibility. 

Line 274: A sentence should not start with a conjunction. 'And' Please revise.

Line 337: grammatical errors, please revise.

Author Response

Response to Reviewer 2 Comments

Dear reviewers,

Thank you for the valuable comments from the reviewers concerning our manuscript entitled “Supplemental enzyme and probiotics on the growth performance and nutrient digestibility of broilers fed with a newly harvested corn diet” (Manuscript ID: animals-1848559). The comments have been highly valuable and helpful for revising and improving our manuscript. We have endeavored to incorporate the feedback and revised our manuscript accordingly, and use the “Track Changes” function for any revisions to the manuscript. The detailed answers to the reviewers are as follows:

Reviewer 2 Comments:

  • Q: I suggest the same parameters be given from Table 1 and 3. The information should be consistent for comparison of the corn (various strains) and the corn diets. Table 3 should include values for CF%, Ash%, soluble penostans. Starch value can also be placed in Table 1 if available.

Answer: Thanks for your advice. Your suggestion has given us a lot of inspiration, but unfortunately we did not consider this issue before, so there is a lack of data in this regard, but fortunately this will not have much impact on our research. We will consider this recommendation in a follow-up study.

  • Q:In text referencing should be in the journal format (please revise throughout the manuscript) eg. Line 156, 319.

Answer: Thanks for your kind advice. We have carefully checked the manuscript reference format and made revisions as required by the journal. Please see L 63, 88, 97, 178-179, 307, 331, 334 and 338-339.

  • Q: In the results section (3.3). Some mention can be made on the effect of the NC diets supplemented with BCC and XL+BCC were comparable with the PC with respect to starch digestibility.

Answer: Thanks for your advice. We have supplemented this in the results section (3.3) based on your suggestion. Please see L 252-254.

  • Q: Line 274: A sentence should not start with a conjunction. 'And' Please revise.

Answer: Thanks for your suggestion. We had revised the writing in our paper as advised. Please see L 292.

  • Q: Line 337: grammatical errors, please revise.

Answer: Thanks for your comment and kind advice. We had already modified the writing. Please see L 358.

Please refer to the attachment for revised manuscript.

Finally, thank you again for your valuable suggestions for this study.

Best regards.

Reviewer 3 Report

Dear Authors,

Thank you for submitting this interesting paper describing enzyme and probiotic supplementation effects on broilers. I found your work to be engaging overall. 

At current however, there are some revisions required in the manuscript to ensure the work is scientifically robust. I have attached the PDF version of the manuscript with specific comments. Additionally, please consider the following points:

1. Proof read. In scientific writing, make sure that species are introduced with their scientific and common name on first mention. Avoid beginning sentences with 'and'. Many sentences are currently confusingly worded.

2. ANOVA is a parametric test and assumes normal distribution of data. Did you test your data for normality and if so what was the finding? Please report it here. If the data are not parametric an alternative test (such as Kruskal Wallis) would be more appropriate.

 3, When using p values, please report that actual p value (not p<0.05) and provide the test statistic too. This is important as it provides information on effect size and how the tests were run.

Author Response

Response to Reviewer 3 Comments

Dear reviewers,

Thank you for the valuable comments from the reviewers concerning our manuscript entitled “Supplemental enzyme and probiotics on the growth performance and nutrient digestibility of broilers fed with a newly harvested corn diet” (Manuscript ID: animals-1848559). The comments have been highly valuable and helpful for revising and improving our manuscript. We have endeavored to incorporate the feedback and revised our manuscript accordingly, and use the “Track Changes” function for any revisions to the manuscript. The detailed answers to the reviewers are as follows:

Reviewer 3 Comments:

  • Q:could improve? Surely this should be past tense?

Answer: Thanks for your advice. We have modified the sentence tense according to your suggestion. Please see L 18.

  • Q:“New grain phenomenon happens in newly harvested corn, which can cause low nutrient digestibility and diarrhea in animals.” Why is this?

Answer: Thanks for your attention. We have added explanations and revised the sentence as: “New grain phenomenon happens in newly harvested corn because of its high content of anti-nutritional factors (ANFs), which can cause low nutrient digestibility and diarrhea in animals.” Please see L 24-25.

  • Q:“A total of 624 Arbor Acres Plus male broiler chickens were randomly divided into 8 treatment groups”. Include scientific name when first mentioning a species.

Answer: Thanks for your suggestion. Arbor Acres Plus broiler mentioned in this article is the scientific name. Please see L 28-29.

  • Q:Describe the implications / practical applications of your study.

Answer: Thanks for your advice. We have added the implications / production applications of this study to the abstract. Please see L 55-57.

  • Q:Some of the keywords are already in the title. Remove these repeats and select new terms to increase article discoverability.

Answer: Thanks for your suggestion. We have removed repeats and selected new terms to add to the keywords. Please see L 58.

  • Q:“can cause harmful bacteria such as Hungatella hathewayi and Bacteroides fragilis in the gut of broiler chickens.” Cause the bacteria to do what?

Answer: Thanks for your comment. We have supplemented this content. Please see L 67-68.

  • Q:“Animal pancreases only secret α-amylase to degrade α-1, 4-glucosidic bonds, while amylopectin can only be hydrolyzed by amylopectase”. Make sure this is cited.

Answer: Thanks for your kind advice. We have added relevant reference to this sentence. Please see L 72 and 379.

  • Q:“Utilizing supplemental enzymes to eliminate the negative effects of anti-nutritional factors of feed ingredients is the common practice.” cite to support this statement.

Answer: Thanks for your kind advice. We have added relevant reference to this sentence. Please see L 79 and 385.

  • Q:“xylanase had a tendency to decrease body weight gain of broilers, and increased the count of Campylobacter, Helicobacter and Butyricicoccus in the ceca.”  

Answer: Thanks for your comment. We have replaced increased with increase. Please see L 102.

  • Q:“If Pediococcus acidilactici BCC-1 individually or combination with enzymes can utilize...”  or in combination.

Answer: Thanks for your comment. We have already modified the writing. Please see L 113.

  • Q:Where were they from? Please deswcribe the conditions in which they were kept as environmental conditions can effect digestibility.

Answer: Thanks for your comment. We have added the source of broiler. Please see L 128-129. Issues concerning broiler rearing conditions are addressed in this article L 150-155.

  • Q:“each group including 6 replicates cages”  replicate.

Answer: Thanks for your comment. We have already modified the writing. Please see L 130.

  • Q:In this format, the treatments are difficult to follow. I recommend reformatting into a table.

Answer: Thanks for your suggestion. We have reformatted the table based on your suggestion, which does give a clearer picture of how the experiment was treated. Please see L 136.

  • Q:“its chemical composition analysis was shown in Table1.”  spacing

Answer: Thanks for your comment. We had already modified the writing. Please see L 138.

  • Q:How old is the normal corn? Are the two harvested from the same area?

Answer: Thanks for your comment. Generally, corn is stored for about a year after it is harvested before being used. In this experiment, corn stored for one year was selected as the normal corn diet. Furthermore, both varieties of corn were harvested in Zhuozhou (Hebei, China).

  • Q:“(Amerah et al., 2007 [19]).” check citation format here

Answer: Thanks for your attention, We have modified the citation format according to the journal requirements. Please see L 161.

  • Q:“Date were analyzed by one-way analysis of variance (ANOVA)”  Data

Answer: Thanks for your comment. We had already modified the writing. Please see L 199.

  • Q:ANOVA is a parametric test and should only be run on normally distributed data? Please state here whether you tested for normal distribution and what the outcome was.

Answer: Thanks for your comment. We have supplemented the relevant content. Please see L 199-211.

  • Q:Please provide the actual p value and test statistic in all cases. State all p values and provide the test statistics throughout the results section.

Answer: Thanks for your suggestion. We have modified the result description part according to your suggestion, but we found that since this study involved more treatment groups, when describing the data, multiple groups of data will be summarized and described, if adding the actual p value will make the whole article look less beautiful and sophisticated. Here, we still choose to describe the p value in the original way, and thank you again for your valuable suggestions for this study.

  • Q:Table 5. What do the letters indicate?

Answer: Thanks for your comment. We have explained what the letters mean below the first table of results. Please see L 227-229.

  • Q:“which in turn maybe lead to the occurrence of diarrhea in poultry” may lead

Answer: Thanks for your comment. We had already modified the writing. Please see L 279.

  • Q:“and feeding the newly harvested corn diet would cause diarrhea in broilers.” this is a very strong statement. Do you have any evidence to support this?

Answer: Thanks for your comment. Our findings are consistent with the findings of Yin et al. (2022), so we came to this conclusion, and we have added this sentence in the article. Please see  L 295.

  • Q:In the conclusion make sure the non-significant results are described here, too

Answer: Thanks for your advice. We have added the corresponding content. Please see L 351-352.

  • Q:Any abbreviations should be followed by a full stop.

Answer: Thanks for your comment. We have revised the reference format as requested. Please see L377-487.

  • Q:Italicise all scientific names here. Also - check whether the species names were capitalised in the original article

Answer: Thanks for your comment. We have revised the reference format as requested. Please see L409 415-416 420 422 431-432 445 459 461-462 472-473 and 480.

  • Q:Explanation of reference page numbers.

Answer: Thanks for your comment. The page numbers of L 435 476 and 487 are from the citation information itself.

Please refer to the attachment for revised manuscript.

Finally, thank you again for your valuable suggestions for this study.

Best regards.

Round 2

Reviewer 1 Report

Thank you very much for your efforts. However, the Ms may need further English editing for smooth reading due to hard reading for some new added statements  i.e 

 L 109-112  As far as we known, Pediococcus acidilactici BCC-1 individually or in combination with enzymes can utilize the high level of soluble sugars in newly harvested corn, and promote broiler growth and improve nutrient digestion and absorption in newly harvested corn diet is unknown. 

L343-344 but didin`t not affect growth performance and di- 343 gestibility of curde protein, crude fiber and starch.

l351-353, Pediococcus acidilactici BCC-1 (109 351 cfu/kg) and protease (800,000 U/g) indicidually or in combination with glucoamylase 352 (800,000 U/g) were supplemented in newly harvested corn diets for growing broilers.

I think can/could be supplemented 

Author Response

Response to Reviewer 1 Comments

Dear reviewers,

Thanks to the reviewers for their positive response concerning our manuscript entitled “Supplemental enzyme and probiotics on the growth performance and nutrient digestibility of broilers fed with a newly harvested corn diet” (Manuscript ID: animals-1848559). We have endeavored to incorporate the feedback and revised our manuscript accordingly, and use the “Track Changes” function for any revisions to the manuscript. The detailed answers to the reviewers are as follows:

  • Q: L 109-112  As far as we known, Pediococcus acidilactici BCC-1 individually or in combination with enzymes can utilize the high level of soluble sugars in newly harvested corn, and promote broiler growth and improve nutrient digestion and absorption in newly harvested corn diet is unknown.

Answer: Thanks for your kind advice. We have revised the statement. Please see L 113-117.

  • Q: L343-344 but didin`t not affect growth performance and digestibility of curde protein, crude fiber and starch.

Answer: Thanks for your kind advice. We have revised the statement. Please see L 352-353.

  • Q: L351-353, Pediococcus acidilactici BCC-1 (109 cfu/kg) and protease (800,000 U/g) indicidually or in combination with glucoamylase (800,000 U/g) were supplemented in newly harvested corn diets for growing broilers.

Answer: Thanks for your kind advice. We have revised the statement. Please see L 361-363.

We thank again for your earnest review.

Best regards.

Reviewer 3 Report

Dear Authors,

Thank you for addressing the majority of concerns raised during the first revision. While most are addressed, one major concern remains. I have added the original concern and your response below:

  • Q:Please provide the actual p value and test statistic in all cases. State all p values and provide the test statistics throughout the results section.

Answer: Thanks for your suggestion. We have modified the result description part according to your suggestion, but we found that since this study involved more treatment groups, when describing the data, multiple groups of data will be summarized and described, if adding the actual p value will make the whole article look less beautiful and sophisticated. Here, we still choose to describe the p value in the original way, and thank you again for your valuable suggestions for this study.

While it is nice that you want your manuscript to look beautiful and sophisticated, the lack of test statistic and exact p values results in poor science and reduces the repeatability of your work. Please amend all statistics as per the original comment.

Author Response

Response to Reviewer 3 Comments

Dear reviewers,

Thanks to the reviewers for their positive response concerning our manuscript entitled “Supplemental enzyme and probiotics on the growth performance and nutrient digestibility of broilers fed with a newly harvested corn diet” (Manuscript ID: animals-1848559). We have endeavored to incorporate the feedback and revised our manuscript accordingly, and use the “Track Changes” function for any revisions to the manuscript. The detailed answers to the reviewers are as follows:

Thanks for your kind advice. We have used exact p values in the Results section as you suggested. Please see 3.Results section.

We thank again for your earnest review.

Best regards.

Round 3

Reviewer 3 Report

Dear Authors,

Thank you for clarifying the points relating to statistical analysis and p values. The paper is now in a better position overall.

Author Response

Dear reviewers,

Thank you for reviewing our manuscript and for the constructive comments, which greatly helped us to improve the manuscript. 

Best regards